# LINC00084/miR-204/ZEB1 Axis Mediates Myofibroblastic Differentiation Activity in Fibrotic Buccal Mucosa Fibroblasts: Therapeutic Target for Oral Submucous Fibrosis

**DOI:** 10.3390/jpm11080707

**Published:** 2021-07-23

**Authors:** Yu-Hsien Lee, Yi-Wen Liao, Ming-Yi Lu, Pei-Ling Hsieh, Cheng-Chia Yu

**Affiliations:** 1School of Dentistry, Chung Shan Medical University, Taichung 40201, Taiwan; cos1018@gmail.com (Y.-H.L.); mylu@csmu.edu.tw (M.-Y.L.); 2Department of Dentistry, Chung Shan Medical University Hospital, Taichung 40201, Taiwan; 3Institute of Oral Sciences, Chung Shan Medical University, Taichung 40201, Taiwan; rabbity0225@gmail.com; 4Department of Medical Research, Chung Shan Medical University Hospital, Taichung 40201, Taiwan; 5Department of Anatomy, School of Medicine, China Medical University, Taichung 404333, Taiwan

**Keywords:** oral submucosal fibrosis, arecoline, LINC00084, miR-204, myofibroblast

## Abstract

Oral submucosal fibrosis (OSF) is a precancerous condition in the oral cavity and areca nut consumption has been regarded as one of the etiologic factors implicated in the development of OSF via persistent activation of buccal mucosal fibroblasts (BMFs). It has been previously reported that an epithelial to mesenchymal transition (EMT) factor, ZEB1, mediated the areca nut-associated myofibroblast transdifferentiation. In the current study, we aimed to elucidate how areca nut affected non-coding RNAs and the subsequent myofibroblast activation via ZEB1. We found that long non-coding RNA LINC00084 was elicited in the BMFs treated with arecoline, a major alkaloid of areca nut, and silencing LINC00084 prevented the arecoline-induced activities (such as collagen gel contraction, migration, and wound healing capacities). The upregulation of LINC00084 was also observed in the OSF tissues and fibrotic BMFs (fBMFs), and positively correlated with several fibrosis factors. Moreover, we showed knockdown of LINC00084 markedly suppressed the myofibroblast features in fBMFs, including myofibroblast phenotypes and marker expression. The results from the luciferase reporter assay confirmed that LINC00084 acted as a sponge of miR-204 and miR-204 inhibited ZEB1 by directly interacting with it. Altogether, these findings suggested that the constant irritation of arecoline may result in upregulation of LINC00084 in BMFs, which increased the ZEB1 expression by sequestering miR-204 to induce myofibroblast transdifferentiation.

## 1. Introduction

As one of the potentially malignant oral disorders, oral submucous fibrosis (OSF) is a chronic inflammatory condition in the oral cavity characterized by the excessive deposition of extracellular matrix (ECM) components, such as collagen. OSF has been known to be associated with the habit of areca nut chewing [1] with a malignant transformation rate of around 5.2–8.6% [2,3,4]. Various studies have shown that an increase in myofibroblasts is implicated in the severity and progression of OSF and oral cancer [5,6]. It has been indicated that the ingredients of areca nut elicit tissue inflammation, myofibroblast differentiation, and alteration of ECM turnover through modulation of numerous molecular signaling such as transforming growth factor-beta 1 (TGF-β1), plasminogen activator inhibitor-1 (PAI-1), tissue inhibitors of metalloproteinases (TIMPs), and metalloproteinases (MMPs) [7]. Additionally, several studies have shown that the areca nut-associated inflammatory cytokines induce cells to undergo epithelial to mesenchymal transition (EMT) [8,9], a potential driving force of myofibroblast transdifferentiation [10]. Moreover, a growing body of research suggests non-coding RNAs play integral roles in the activation of myofibroblasts. 

Non-coding RNAs (ncRNAs) refer to RNAs that do not encode proteins and are divided into various categories, including housekeeping ncRNAs (such as ribosomal RNA) and regulatory ncRNAs (such as microRNAs and long non-coding RNAs) [11]. MicroRNAs (miRNAs) are approximately 19–23 nucleotides in length and can bind to target mRNAs at the 3′ untranslated region (3′-UTR) to regulate their expression. Long non-coding RNAs (lncRNAs) are mRNA-like transcripts (>200 nucleotides in length) lacking significant open reading frames and can modulate gene expression through various modes of action, such as serving as a scaffold for chromatin-modifying complexes, a decoy of mRNAs or a sponge of miRNAs (see Review [12]). Emerging evidence reveals that numerous biological processes are orchestrated by the interplay between miRNAs and lncRNAs, including activities involved in the development of OSF [13]. Nevertheless, the detailed mechanisms underlying the direct and indirect epigenetic regulation in myofibroblast activation by miRNAs and lncRNAs remain largely unknown.

In the present study, we tested the effect of arecoline (an alkaloid from areca nut) on the expression of lncRNAs LINC00084 in buccal mucosal fibroblasts (BMFs) and verified our findings in clinical samples. Subsequently, we examined whether silencing LINC00084 affected the myofibroblast activities and the downstream signaling in an effort to delineate possible molecular events in response to areca nut stimulation.

## 2. Materials and Methods

### 2.1. Cell Culture

All procedures were conducted under the approval from the Institutional Review Board of Chung Shan Medical University Hospital, Taichung, Taiwan (approval number: CSMUH No. CS18124). OSF tissues were collected after obtaining written informed consent. Primary cultures, including BMFs and fBMFs, were cultivated as previously described and the third and eighth passages will be used in the OSF study. Cells were passaged routinely at 90% confluence. BMFs or fBMFs were migrated from the tissue margin and began to proliferate in a dish containing 10% fetal bovine serum in a DMEM medium [13].

### 2.2. Reagents

Arecoline and collagen solution from the bovine skin were purchased from Sigma–Aldrich (St. Louis, MO, USA). Arecoline was used to induce myofibroblast activities in BMFs. 

### 2.3. Tissue Acquisition

All procedures performed in this study involving human participants were in accordance with the tenets of the Declaration of Helsinki and were approved by the Institutional Review Committee at Chung Shan Medical University (approval number: CSMUH No. CS18124), Taichung, Taiwan. Histological normal or fibrotic mucosa tissues were retrieved from normal subjects or OSF patients recruited in the Department of Dentistry, Chung Shan Medical University Hospital. Tissues were collected after obtaining written informed consent. RNA was extracted from these tissues then used for quantitative real-time PCR analysis. 

### 2.4. Quantitative Real-Time PCR and Western Blot Analysis

These analyses were performed to detect the expression of LINC00084, ACTA2, ZEB1, and GAPDH. All procedures were conducted as previously described [10]. The primer sequences used in this study were listed as follows: LINC00084: 5′- GACCTCTCACCTACCCACCT -3′ and 5′- CTTGTACCCTCCCAGCGTTT -3′; and GAPDH: 5′-CTCATGACCACAGTCCATGC-3′ and 5′- TTCAGCTCTGGGATGACCTT-3′.

### 2.5. Lentiviral-Mediated Knockdown or Overexpression of LINC00084

Lentivirus production followed previously described protocols. The target sequences for LINC00312 were listed as follows: Sh-LINC00084-1: 5′-AAAAGGAAGTGAGAAGTTGCTTATTGGATCCAATAAGCAACTTCTCACTTCC-3′; Sh-LINC00084-2: 5′-AAAAGGACTACTTGGCAACTTTATTGGATCCAATAAAGTTGCCAAGTAGTCC-3′ [13].

### 2.6. Collagen Gel Contraction

Cells were suspended in collagen gel solution (Sigma–Aldrich, St. Louis, MO, USA) and added into a 24-well-plate followed by incubation at 37 °C for 2 h. After polymerization, the gels were further incubated within 0.5 mL medium for 48 h. The collagen gel size change (contraction index) was quantified using ImageJ software (NIH, Bethesda, MD, USA) [10].

### 2.7. Cell Migration and Invasion Assays

1 × 10^5^ cells in a medium with low serum were added into the upper chamber of a transwell (Corning, Acton, MA, USA) and medium supplemented with higher serum were used as a chemoattractant in the lower chamber followed by 24 h incubation. Cells on the lower surface of the insert membrane were stained with crystal violet. The number of migration cells in a total of five randomly selected fields was measured [13].

### 2.8. Dual-Luciferase Reporter Gene Assay

To detect the connection of LINC00084 and miR-204, LINC00084 3′-untranslated region (3′-UTR) wild-type (Wt) and mutant type (Mut) luciferase reporter plasmids were constructed. Subsequently, two plasmids were respectively mixed with miR-204 plasmids and NC plasmids and then the mixtures were respectively co-transfected into cells [13].

### 2.9. RIP (RNA Immunoprecipitation) Assay

RIP experiments were performed using Magna RIP kit (Millipore, Billerica, MA, USA). The cell lysate was incubated with RIP immunoprecipitation buffer, which contained magnetic beads conjugated with Ago2 antibody (Abcam, Burlingame, CA, USA) and NC IgG (Abcam, Cambridge, MA, USA). qRT-PCR analysis was conducted after co-precipitate.

### 2.10. Statistical Analysis

Three replicates of each experiment were performed. Statistical calculations of the data were performed using Student’s t-test for the differences between two independent samples and one-way analysis of variance (ANOVA) with post hoc Scheffe and Tukey’s tests for more than two groups with unequal and equal sample sizes per group, respectively. A *p*-value < 0.05 was considered to be statistically significant. All statistical analyses were performed using IBM SPSS Statistics (SPSS 27 for Windows).

## 3. Results

### 3.1. Arecoline-Induced LINC00084 Promotes Myofibroblast Activation

It has been known that the activated myofibroblasts migrate to the site of injury and are responsible for the wound closure as well as the secretion of ECM components during the process of healing and fibrosis [14]. Besides, arecoline has been proven to be sufficient to elicit the myofibroblast transdifferentiation of BMFs [15]. To determine whether LINC00084 was implicated in the arecoline-induced myofibroblast activation, we examined the relative expression of LINC00084 in two normal BMFs following treatment of various concentrations of arecoline. As shown in Figure 1A, arecoline dose-dependently enhanced the expression of LINC00084 in two BMFs. Subsequently, we tested if the upregulation of LINC00084 contributed to myofibroblast activation by conducting the short hairpin RNA (shRNA)-mediated knockdown experiment. We observed that the arecoline-stimulated collagen gel contractility (Figure 1B), transwell migration (Figure 1C), and wound healing (Figure 1D) abilities were all abrogated in two BMFs transfected with sh-LINC00084, indicating that loss of LINC00084 disturbs the arecoline-provoked myofibroblast activation and LINC00084 may participate in the pathogenesis of areca nut-associated OSF.

### 3.2. The Upregulated LINC00084 in OSF Tissues Is Positively Associated with Several Fibrosis Markers

From a heatmap of RNA-sequencing result, we showed LINC00084 was aberrantly overexpressed in the OSF samples compared to normal specimens (Figure 2A). To validate this finding, we conducted the qRT-PCR and confirmed that LINC00084 was upregulated in OSF tissues (Figure 2B). Previously, it has been shown that TGF-β [9], interleukin-6 (IL-6) [8], and transglutaminase 2 (TGM2) [16] were all related to the areca nut-induced activation of oral fibroblasts. Here, we found that LINC00084 was positively correlated with these three factors (Figure 2C–E) using the oral cancer data from the cancer genome atlas (TCGA), suggesting that higher expression of LINC00084 may be associated with the progression of this precancerous OSF. Moreover, we showed that LINC00084 was indeed upregulated in the fibrotic BMFs (fBMFs) derived from OSF tissues (Figure 2F).

### 3.3. Silencing LINC00084 in fBMFs Mitigates the Myofibroblasts Features

To examine the significance of LINC00084 in the maintenance of myofibroblast features, we employed shRNAs to knockdown of the expression of LINC00084 in fBMFs (Figure 3A). As expected, silencing of LINC00084 reduced the collagen gel contraction capacity of fBMFs (Figure 3B). On the other hand, upregulation of alpha-smooth muscle actin (α-SMA) has been shown to induce contractile activity and be used as an indication of myofibroblast activation [17]. Additionally, it has been revealed that the arecoline-induced collagen contraction of BMFs is mediated by the direct binding of EMT inducer, zinc finger E-box binding homeobox 1 (ZEB1) to the α-SMA promoter [10]. We observed that the expression levels of ZEB1 and α-SMA were both suppressed in fBMFs with sh-LINC00084 (Figure 3C). Similarly, the migration (Figure 3D) and wound healing (Figure 3E) capacities were markedly inhibited in fBMFs following repression of LINC00084. These findings indicate that downregulation of LINC00084 may be a promising approach to diminish the persistent activation of fBMFs.

### 3.4. LINC00084 Acts as a Molecular Sponge of miR-204

In terms of the subcellular location of LINC00084, we found it was distributed in both the nucleus and cytoplasm, and primarily localized in the cytoplasm of fBMFs (Figure 4A). The potential binding sites between LINC00084 and miR-204 were presented in Figure 4B, and the luciferase activity was shown in Figure 4C. We observed that transfection of miR-204 significantly downregulated the luciferase activity of the plasmids expressing wild-type (Wt) LINC00084 in fBMF, while the activity was not affected in which the LINC00084 was mutagenized (Mut) (Figure 4C). In fBMFs, we showed downregulation of LINC00084 upregulated the expression of miR-204 (Figure 4D), and overexpression of LINC00084 repressed the expression of miR-204 in normal BMFs (Figure 4E). To probe the presence of LINC00084 in the RNA-induced silencing complex (RISC), RNA RIP experiments were performed using anti-Ago2 in fBMFs and we showed that both LINC00084 and miR-204 could be precipitated by Ago2 antibody (Figure 4F,G). Moreover, we found that the expression of LINC00084 was suppressed when the miR-204 inhibitor was co-existent (Figure 4G), indicating that LINC00084 interacted with miR-204 in Ago2-dependent manner.

### 3.5. MiR-204 Regulates the Myofibroblast Activities by Directly Targeting ZEB1

Given that ZEB1 is crucial to the transdifferentiation of myofibroblast [10] and a predicted target of miR-204, ZEB1 wild-type (Wt) or mutant 3′-UTR (Mut) (Figure 5A) was subcloned into a luciferase reporter vector and co-transfected with miR-204 mimic or miR-scrambled (negative control). The luciferase activity of each combination was assessed and transfection of miR-421 significantly inhibited the activity of the wild-type group (Figure 5B). Besides, results from western blot demonstrated that the expression of ZEB1 was inhibited in fBMFs with miR-204 mimics (Figure 5C). Next, we showed that two myofibroblast features, the increased collagen gel contractility, and transwell migration, were both attenuated in fBMFs transfected with miR-204 mimics (Figure 5D,E). Taken together, our findings in conjunction with the results from the previous investigation [10] suggested that miR-204 hampers the myofibroblast activation via suppression of ZEB1.

### 3.6. LINC00084 Mediates the Myofibroblast Activation via Sponge Regulation of miR-204-ZEB1 Interaction

To substantiate LINC00084 affects myofibroblast activities through modulation of the miR-204-ZEB1 axis, various characteristics of myofibroblast were tested. First, the expression of ZEB1 and α-SMA in two BMFs with forced expression of LINC00084 or miR-204 were examined. As shown in Figure 6A, ectopic expression of miR-204 abrogated the upregulation of ZEB1 and α-SMA in LINC00084-overexpressing BMFs. Similar results were observed using transwell migration and collagen gel contraction assays. We showed that overexpression of miR-204 abolished these two features induced by LINC00084 elevation (Figure 6B,C), suggesting that miR-204 plays an essential role in the LINC00084-associated transdifferentiation of myofibroblasts. Furthermore, our results showed that the increased expression level of α-SMA in the LINC00084-transfected cells were also prohibited after silencing ZEB1 (Figure 6D). Likewise, the enhanced transwell migration ability and collagen gel contractility in LINC00084-overexpressing BMFs were repressed following the knockdown of ZEB1 (Figure 6E,F). Collectively, these results depicted that LINC00084 regulates the myofibroblast hallmarks via competing with miR-204, thereby increasing ZEB1 and leading to OSF.

## 4. Discussion

LINC00084 (nuclear enriched autosomal transcript 1; NEAT1) is a nuclear-retained lncRNA and was identified by Hutchinson et al. using nuclear and cytoplasmic RNA fractions from human fibroblasts and lymphoblasts [18]. LINC00084 normally resides in paraspeckles, but disassociates from the nuclear bodies to translocate into the cytoplasm upon stimulation with inflammasome-activating signals [19]. It has been shown that the expression of LINC00084 is elevated in response to stress [20], and promotes various types of fibrosis diseases. For instance, it has been revealed that the expression of LINC00084 was significantly increased in CCl4-induced liver fibrosis mice. Yu et al. showed that LINC00084 directly interacted with miR-122, which hindered its inhibition of Kruppel-like factor 6 [21]. In diabetic nephropathy, LINC00084 exhibited a fibrosis effect to increase the expression of TGF-β1, fibronectin, and collagen IV in the glucose-induced mouse mesangial cells via activating Akt/mTOR signaling [22]. LINC00084 also has been found to be secreted by cardiomyocytes via vesicles to the neighboring fibroblasts under hypoxic conditions. It has been demonstrated that LINC00084 was indispensable for several fibrosis features, such as migration ability of myofibroblasts and expression of fibrotic genes [23]. Evidence also suggested that LINC00084 was involved in the progression of oral cancer as it was highly elevated in the oral premalignant lesions [24] and found to accelerate cell proliferation and invasion of oral cancer cells by regulating miR-365/RGS20 signaling [25]. In line with these results, we showed that LINC00084 was upregulated in the OSF tissues and myofibroblasts derived from OSF specimens. Our results demonstrate that the aberrantly overexpression of LINC00084 in OSF was due to the stimulation of the areca nut. Moreover, we demonstrated that LINC00084 was implicated in the development of arecoline-associated OSF through interfering with the miR-204/ZEB1 axis.

MiR-204 has been shown to protect tubular epithelial cells from chronic fibrotic change after ischemia-reperfusion injury via targeting SP1 and regulation of EMT in the tubular epithelial cells [26]. Upregulation of miR-204 also inhibited the proliferation, migration, invasion of TGF-β1-treated human tenon capsule fibroblasts as well as the expression of EMT and fibrosis markers [27]. Aside from its anti-fibrosis effect, miR-204 was found to suppress ZEB1 to regulate EMT phenotype and radioresistance of nasopharyngeal carcinoma cells [28]. Lu et al. demonstrated that there was reciprocal repression between LINC00084 and miR-204, which was consistent with our findings. We showed that miR-204 mimics downregulated myofibroblast activities in fBMFs and attenuated the LINC00084-induced myofibroblast activation. Our results indicate that arecoline upregulated the expression of LINC00084, which sequestered the suppressive effect of miR-204 on ZEB1 and resulted in oral fibrogenesis. ZEB1 has been revealed to implicate in the arecoline-induced collagen contraction of BMFs and other myofibroblast features via direct binding to the α-SMA promoter [10]. We showed that ZEB1 participated in the LINC00084-mediated myofibroblast activation as well. 

In summary, our results indicate that the arecoline-stimulated LINC00084 functions as an miRNA sponge of miR-204 to upregulate ZEB1 expression and initiate myofibroblast transdifferentiation (Figure 7). Additionally, we showed that suppression of LINC00084 may be a viable strategy to downregulate the fibrosis features of fBMFs and ameliorate the progression of OSF. 

## 5. Conclusions

Our findings demonstrate a potential molecular mechanism underlying the pathogenesis of areca nut-associated OSF. The elevated LINC00084 enhances the myofibroblasts activation by upregulation of ZEB1 through sponging miR-204, which may lead to an increase of ECM components and fibrosis. As such, targeting LINC00084 by drugs or natural compounds may be employed in the clinical setting to prevent the exacerbation of OSF.

## Figures and Tables

**Figure 1 jpm-11-00707-f001:**
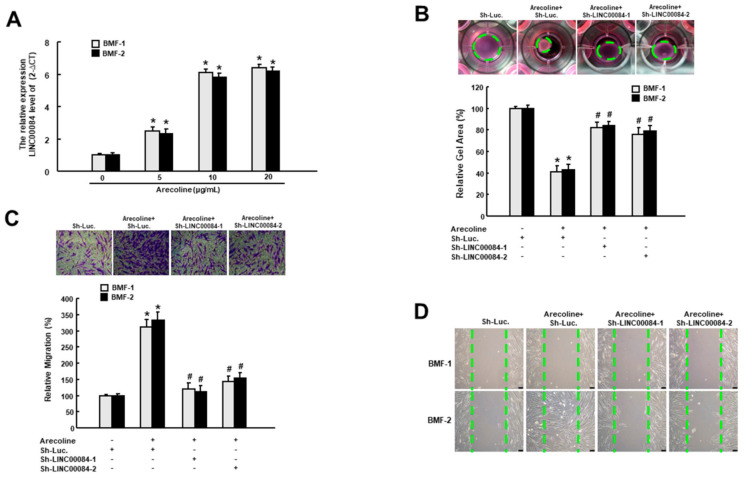
The arecoline-elicited LINC00084 mediates the myofibroblast activation. (**A**) The relative gene expression of LINC00084 in two buccal mucosal fibroblasts (BMFs) in response to the indicated concentration of arecoline. * *p* < 0.05 compared to control group (no arecoline); the arecoline-induced myofibroblast activities, including the enhanced collagen gel contraction (**B**), transwell migration (**C**), and wound healing capacities (**D**) were measured in arecoline-treated BMFs transfected with sh-LINC00084. Results are means ± SD of triplicate samples from three experiments. * *p* < 0.05 compared to Sh-Luc. group. # *p* < 0.05 compared to the arecoline + Sh-Luc. group.

**Figure 2 jpm-11-00707-f002:**
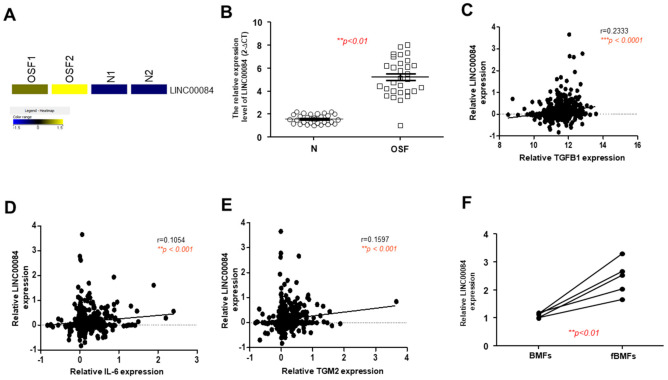
The expression of LINC00084 is aberrantly upregulated in OSF tissues and patient-derived BMFs. (**A**) A heatmap showing LINC00084 was highly expressed in two OSF and normal buccal mucosal tissues; (**B**) the relative expression of LINC00084 in normal and OSF specimens (*n* = 30); analysis of the relationship between LINC00084 and various OSF-associated factor, including TGF-β (**C**), IL-6 (**D**), and TGM2 (**E**) using OSCC data from the cancer genome atlas (TCGA); and (**F**) the relative expression of LINC00084 in normal human BMFs and fibrotic BMFs (fBMFs) derived from OSF tissues. *** p* < 0.01; **** p* < 0.001.

**Figure 3 jpm-11-00707-f003:**
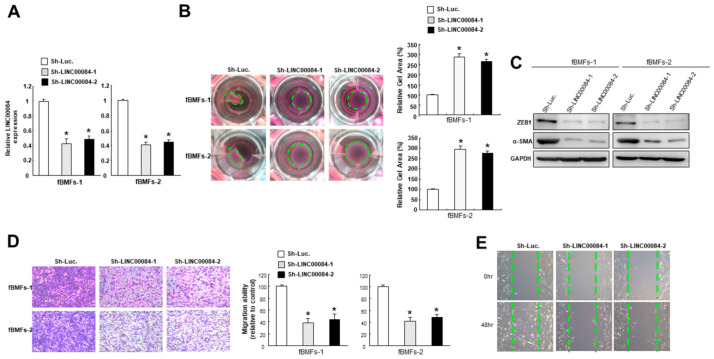
Suppression of LINC00084 attenuates the myofibroblasts properties. (**A**) The knockdown efficiency of sh-LINC00084 in two patient-derived fBMFs; collagen gel contractility (**B**), the expression level of ZEB1 and α-SMA (**C**), transwell migration (**D**), and wound healing (**E**) abilities were examined in two fBMFs transfected with sh-Luc. or sh-LINC00084. Results are means ± SD of triplicate samples from three experiments. * *p* < 0.05 compared to sh-Luc. group.

**Figure 4 jpm-11-00707-f004:**
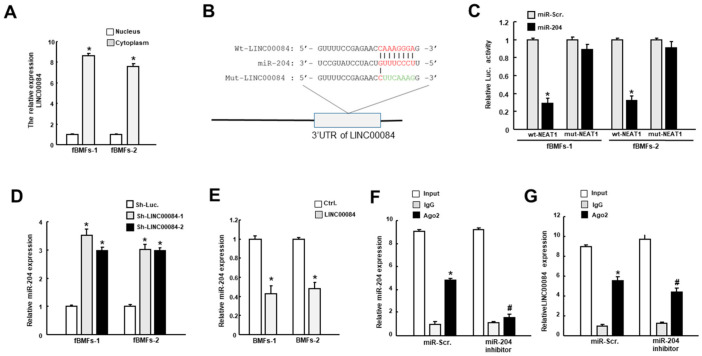
LINC00084 curbs the expression of miR-204 in fBMFs. (**A**) The relative cytoplasmic and nuclear levels of LINC00084 in fBMFs analyzed by qRT-PCR. * *p* < 0.05 compared to the nucleus group; (**B**) schematic representation of the alignment of the LINC00084 base pairing with miR-204. Wild-type (Wt) and mutated (Mut) LINC00084 reporter plasmids were co-transfected with miR-204 or empty vectors (miR-Scr.); (**C**) the relative luciferase activity of each combination in two fBMFs was assessed and only WT reporter activity was suppressed by miR-204; (**D**) the relative expression of miR-204 in two fBMFs transfected with sh-Luc. or sh- LINC00084; (**E**) the expression level of miR-204 in two LINC00084-overexpressing fBMFs; (**F**) enrichment of miR-204 (**F**) or LINC00084 (**G**) relative to input was assessed via the RIP experiment using anti-Ago2 in fBMFs transfected with miR-Scr. or miR-204 inhibitor with anti-IgG as a negative control. The relative expression levels of miR-204 and LINC00084 were detected by qPCR * *p* < 0.05 compared to sh-Luc. or control group; # *p* < 0.05 compared to miR-Scr. group.

**Figure 5 jpm-11-00707-f005:**
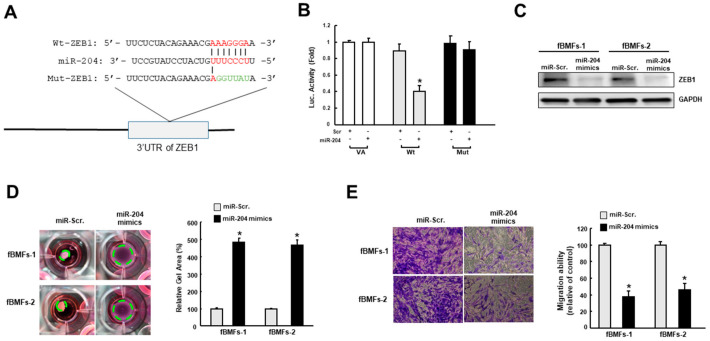
MiR-204 directly binds to the 3′-untranslated region (3′-UTR). (**A**) Schematic presentation of the constructed 3′-UTR plasmids of wild-type (Wt) and mutated (Mut) ZEB1; (**B**) the luciferase activity of each combination was measured and only WT reporter activity was inhibited by miR-204; the protein expression of ZEB1(**C**), collagen gel contractility (**D**), and migration ability (**E**) in two fBMFs transfected with miR-Scr. or miR-204 mimics were examined. * *p* < 0.05 compared to miR-Scr. group.

**Figure 6 jpm-11-00707-f006:**
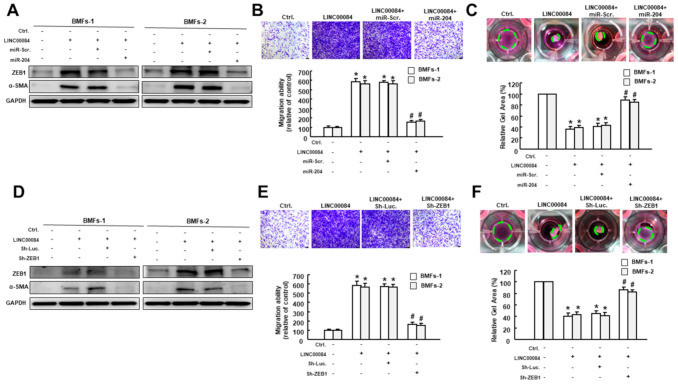
LINC00084 promotes myofibroblast transdifferentiation of BMFs via regulation of miR-204/ZEB1 axis. (**A**) Protein expression of ZEB1 and α-SMA, (**B**) migration ability and (**C**) collagen gel contraction in BMFs co-transfected with LINC00084 overexpression plasmids, miR-Scr., or miR-204 mimics; (**D**) protein expression of ZEB1 and α-SMA, (**E**) migration ability, and (**F**) collagen gel contraction in BMFs co-transfected with LINC00084 overexpression plasmids, Sh-Luc., or Sh-ZEB1. Results are means ± SD of triplicate samples from three experiments. * *p* < 0.05 compared to Ctrl. group. # *p* < 0.05 compared to the LINC00084 overexpression group.

**Figure 7 jpm-11-00707-f007:**
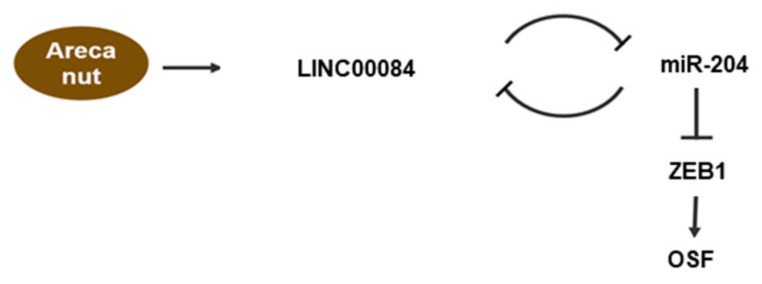
Schematic diagram of the role of LINC00084/miR-204/ZEB1 axis in OSF progression. Arecoline, a major alkaloid from the areca nut, activates the transdifferentiation of buccal mucosal fibroblasts (BMF) into myofibroblasts (including the increased phenotypes and fibrosis markers) via upregulation of LINC00084. LINC00084 could modulate the expression of ZEB1 by acting as a competing endogenous RNA (ceRNA) of miR-204.

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
