# Peer review of "LINC00084/miR-204/ZEB1 Axis Mediates Myofibroblastic Differentiation Activity in Fibrotic Buccal Mucosa Fibroblasts: Therapeutic Target for Oral Submucous Fibrosis"

_jpm, 2021, doi:10.3390/jpm11080707_

Round 1
Reviewer 1 Report
Dear authors,
Your research is well conducted and reporterd, however, I have minor concerns regarding the paper:
-page 2, line 56: adjust the citation;
-More details on the first 3 points of the Materials section would improve the quality of the manuscript and its understandability
-The conclusion are missing, and the clinical relevance of your fidings should be highglighted.
Author Response
Reviewer 1
Dear authors,
Your research is well conducted and reporterd, however, I have minor concerns regarding the paper:
Response: We thank the reviewer for acknowledging our work, and understanding the considerable quantity of data we present herein.
1.-page 2, line 56: adjust the citation;
Response: Thank you for the suggestion. We have adjusted it.
2.-More details on the first 3 points of the Materials section would improve the quality of the manuscript and its understandability
Response: Thank you for the suggestion. We have added the details in the Material and Methods section.
3.-The conclusion are missing, and the clinical relevance of your findings should be highglighted.
Response: Thank you for the suggestion. We have added the conclusion section and mentioned the clinical application in the conclusion.
Reviewer 2 Report
Congrats to the authors on the idea the research is very interesting. However, the text needs clarification and a few corrections.
L80 - 81 - Were the subjects informed that they could opt out of the study at any time ?
Did the subjects give written consent to the study ?
If so, add this information in this part.
L115 - Expand the shortcut as you use it for the first time.
L119 - "Statistical Analysis" - Write what program you used for statistical analysis.
L121 - "Data were expressed as the 120mean ± SD and analyzed by Student's t-test."
Why did you choose this test ? Has the normal distribution been checked? If so write down what tests you used. Please describe this part better.
L124 - I would ask you to add a table with the results obtained, exact "p" values and test values. Would also advise adding the results of the power of statistical tests. This will make the results more readable.
L146 - Whenever you state whether a result was statistically significant or not, add a p value. For example, this provision is insufficient for me - " * p<0.05 compared to Sh-Luc. group. #p<0.05 compared to the arecoline + 146Sh-Luc. group." My comment applies to the entire text.
L304 - 305 - Delete: "For research articles with several authors, a short paragraph specifying their individual contributions must be provided. The following statements should be used"
L305 - Remove the quotation marks.
L312 - Remove triple spaces.
Author Response
Reviewer 2
Comments and Suggestions for Authors
Congrats to the authors on the idea the research is very interesting. However, the text needs clarification and a few corrections.
Response: We thank the reviewer for acknowledging our work, and understanding the considerable quantity of data we present herein.
- L80 - 81 - Were the subjects informed that they could opt out of the study at any time ?Did the subjects give written consent to the study ?If so, add this information in this part.
Response: Thank you for the suggestion. We have included the description regarding the consents.
- L115 - Expand the shortcut as you use it for the first time.
Response: Thank you for the suggestion. We have included the description regarding the consents.
- L119 - "Statistical Analysis" - Write what program you used for statistical analysis.
Response: Thank you for the suggestion. We have mentioned the analysis program in the method section.
- L121 - "Data were expressed as the 120mean ± SD and analyzed by Student's t-test."Why did you choose this test ? Has the normal distribution been checked? If so write down what tests you used. Please describe this part better.
Response: We thank the reviewer's critical comment. Following your advice, we have rephrased the details of Statistical Analysis. Please see the revised manuscript.
- L124 - I would ask you to add a table with the results obtained, exact "p" values and test values. Would also advise adding the results of the power of statistical tests. This will make the results more readable.
Response: We thank the reviewer's critical comment. Following your advice, we have rephrased the details of Statistical Analysis. Please see the revised manuscript.
- L146 - Whenever you state whether a result was statistically significant or not, add a p value. For example, this provision is insufficient for me - " * p<0.05 compared to Sh-Luc. group. #p<0.05 compared to the arecoline + 146Sh-Luc. group." My comment applies to the entire text.
Response: We thank the reviewer's critical comment. Following your advice, we have rephrased the details of Statistical Analysis and added the p value in entire figure legends. Please see the revised manuscript.
- L304 - 305 - Delete: "For research articles with several authors, a short paragraph specifying their individual contributions must be provided. The following statements should be used"
Response: Thank you for the suggestion. We have deleted it.
- L305 - Remove the quotation marks.
Response: Thank you for the suggestion. We have deleted it.
- L312 - Remove triple spaces.
Response: Thank you for the suggestion. We have removed it.
Round 2
Reviewer 2 Report
My last minor comment
L 129/L249/ L300 - Add periods at the end of sentences.